# Exploration of Patient-Derived Pancreatic Ductal Adenocarcinoma Ex Vivo Tissue for Treatment Response

**DOI:** 10.3390/antiox12010167

**Published:** 2023-01-11

**Authors:** Tímea Szekerczés, Arun Kumar Selvam, Carlos Fernández Moro, Soledad Pouso Elduayen, Joakim Dillner, Mikael Björnstedt, Mehran Ghaderi

**Affiliations:** 1Division of Pathology, Department of Laboratory Medicine, Karolinska Institutet, 141 52 Huddinge, Sweden; 2Department of Clinical Pathology and Cancer Diagnostics, Karolinska University Hospital, 141 86 Stockholm, Sweden

**Keywords:** patient-derived PDAC explants, AHR pathway, indole-3-pyruvic acid

## Abstract

Patient-derived tissue culture models are valuable tools to investigate drug effects and targeted treatment approaches. Resected tumor slices cultured ex vivo have recently gained interest in precision medicine, since they reflect the complex microenvironment of cancer tissue. In this study, we examined the treatment response to an internally developed ex vivo tissue culture model from pancreatic ductal adenocarcinoma (PDAC) and in vitro analysis. Seven PDAC tissues were cultured and subsequently treated with indole-3-pyruvic acid (IPA). IPA, which is known as an agonist of the aryl hydrocarbon receptor (AHR) pathway, has antioxidant properties. Genome-wide transcriptome sequencing analysis revealed activation of AHR pathway genes (CYP1A1 and CYP1B1, *p* ≤ 0.05). Additionally, significant upregulation of AHR repressor genes AHRR and TiPARP was also observed (*p* ≤ 0.05), which is indicative of the negative feedback loop activation of AHR pathway signaling. The overall transcriptomic response to IPA indicated that the tissues are biologically active and respond accordingly to exogenous treatment. Cell culture analysis confirmed the significant induction of selected AHR genes by IPA. A morphological examination of the paraffin-embedded formalin-fixed tissue did not show obvious signs of IPA treatment related to tumor cell damage. This study is a proof of concept that ex vivo patient-derived tissue models offer a valuable tool in precision medicine to monitor the effect of personalized treatments.

## 1. Introduction

Pancreatic ductal adenocarcinoma (PDAC) is one of the most devastating cancers, the seventh-leading cause worldwide and the fourth-leading cause in Western countries of cancer-related deaths. Its incidence is increasing alarmingly, and the tumor recurrence rate after resection is still disappointing [1,2]. Early diagnosis of PDAC is almost impossible due to its complex etiology, increased risk factors (e.g., obesity, chronic pancreatitis, diabetes mellitus, or family history of genetic predisposing factors (BRCA1, BRCA2, PALB1 and ATM)), and lack of specific symptoms [1]. Thus, at the time of diagnosis, most patients suffer from a disseminated and/or locally advanced tumor, which usually excludes the possibility of less invasive surgical resection of the tumor. In patients with advanced PDAC, the most common first-line treatment is FOLFIRINOX (i.e., a combination of 5-Fluorouracil, Leucovorin, Irinotecan, and Oxaliplatin) or Gemcitabine plus Nab-Paclitaxel. However, despite being the best available and most widely prescribed medications, they are less likely to significantly improve the 5-year survival rate of PDAC, which is lower than 9% [1,3]. One challenge in developing an efficient drug to treat pancreatic cancer is making the compound accessible to the tumor cells surrounded by fibrotic, desmoplastic stroma, which is difficult to fully recapitulate in preclinical models [4,5,6].

The existing biological systems that facilitate the discovery of molecular targets, for new therapies and the preclinical evaluation of drug candidates, have certain limitations: they cannot recapitulate the complex structure of tumors nor the important interaction between tumor cells and their microenvironment [7].

Currently, human xenograft mouse models and genetically modified animals are the gold standard for in vivo drug testing and are appropriate assets to study the biology of complex human diseases. Preclinical efficacy trial and toxicology testing can be done using animal models before any treatment in humans [8]. However, the timeline required for production is not realistic for precision medicine initiatives, where the goal is to assess therapeutic efficacy in real-time to aid clinical decision making [7]. During the past decade, a growing body of literature has used ex vivo slice culture models to address key questions related to oncogenic signaling pathways, drug-sensitivity testing, and immunotherapy in different tumor types [9]. The major advantages of patient-derived tissue over other models involve their cost-effectiveness, response projection time, and feasibility to test several drugs in parallel. Patient tumor cultures can be maintained during a routine pathological examination to measure the response to various anti-cancer drugs. Patient-derived tumor explants are metabolically active and contain stromal and immune cells, preserving their native cellular interactions, as they do not involve chemical, enzymatic, or mechanical digestion of PDAC tissue, thus avoiding artificial skewing of cell populations [10]. We previously studied the stability of ex vivo precision cut tissue slices of pancreatic ductal adenocarcinoma and we found that explanted tissue stayed successfully alive ex vivo for at least 4 days. We also analyzed the effect of culturing conditions and how it might impact the global gene expression pattern. Employing genome-wide transcriptome profiling of both parent and explanted tumors, we found that only a small number of genes showed significant expression change during the culturing period. In our previous study, transcriptome analysis was considered to be a valuable tool, complement to morphology for evaluation of ex vivo cultures of PDAC. In the previous study, we also found that the vascular endothelial growth factor A (VEGFA) was significantly upregulated during the entire culture period. Pathway over-representation analysis suggested that VEGFA could participate in HIF-1-derived cell apoptosis via NF-κB and the AP-1 activating factor. We speculated that the stability of the ex vivo tissue functionality is substantially dependent on availability of oxygen [11]. Still, patient-derived explants require optimized culture conditions such as adequate oxygen levels and a balanced concentration of nutrients. The lack of a functional vascular system has been suggested to be the reason for the limited viability of explanted ex vivo models [9,12]. Therefore, further testing is warranted to validate the use of explant models in drug-development processes.

In the present study, we aimed to investigate the reliability of PDAC ex vivo explants and to prove the hypothesis that ex vivo tissue cultures of patients can respond appropriately to treatments and, hence, be considered a suitable tool for studying drug effects in precision medicine at both the morphology and molecular signaling levels. In the current study, we used a PDAC ex vivo tissue and tested whether this is suitable for the screening of drug effects. PDAC tissues were treated with a well-known aryl hydrocarbon receptor (AHR) pathway activator [13], indole-3-pyruvic acid (IPA), and all samples showed significant upregulation of the genes involved in the AHR signaling pathway.

## 2. Materials and Methods

### 2.1. Ethical Approval and Analysis of Patient-Derived Samples

The permission for the study and sample collection was obtained by the Regional Ethical Review Board, Stockholm/Etikprövningsmyndigheten (decision numbers 2012/1657-31/4, 2018-2654/32, and 2019-00788). Written informed consent was provided by all patients. Additionally, the study and the collection of patient samples was based on the ethical guidelines of the Declaration of Helsinki of 1975 revised in 2013, and the samples were completely pseudonymized before being accessed by the authors.

The primary PDAC tumor samples (*n* = 7) for the project were collected from chemotherapy-naive patients whose surgeries performed at the Karolinska University Hospital between January and November 2020. Clinicopathological characteristics are summarized in Table 1.

Patient and tissue samples were processed as described in our previous article [14]. In brief, after the surgical removal the specialist pancreatic pathologist sampled a piece of the tumor from fresh resection specimens. The piece of PDAC tissues were immadietly cut into 350 μm thick slices using a vibrating-blade microtome (VT1200S, Leica Biosystems, Nussloch, Germany), and the first slice was immediately fixed in formalin and embedded in paraffin for histomorphology evaluation, as it was considered the 0-h control. The following tissue slices were placed on an insert (0.4 μm pore size, 30 mm diameter, Millicell®, Millipore, Ireland) in a 35 mm culture dishes (BD Falcon, Thermo Fisher Inc., Sweden). During the process, the tissues were stored on ice [14]. To maintain and treat tissue following the procedure and conditions previously described [14], the slices were first cultured for 24 h under normoxic condition (21% O_2_) at 37 °C and were subsequently treated in duplicate with the tryptophan metabolite “indole-3-pyruvic acid” at 200 μM concentration during subsequent 48 h. The CMRL medium (no glutamine, Gibco, Thermo Fisher Scientific Inc., Waltham, MA, USA) supplemented with 25 mmol/L HEPES (Gibco, Thermo Fisher Scientific Inc., Waltham, MA, USA), 1 mmol/L Sodium pyruvate (Gibco, Thermo Fisher Scientific Inc., Waltham, MA, USA), 3 nmol/L Zinc sulfate (ZnSO_4_, Sigma-Aldrich Chemical Company, St. Louis, MI, USA), 1× Insulin-transferrin-sodium selenite solution (Gibco, Thermo Fisher Scientific Inc., Waltham, MA, USA), 2.5% Human serum (H3667, Sigma-Aldrich Chemical Company, St. Louis, MI, USA), 1× Penicillin-Streptomycin (Gibco, Thermo Fisher Scientific Inc., Waltham, MA, USA), and 100 nmol/L diphenyl di-selenide (Sigma-Aldrich Chemical Company, St. Louis, MI, USA) was used to treat and maintain the tissue pieces. After the 48-h treatment, the tissue slices were fixed in formalin and embedded in paraffin blocks and then 4 μm thick sections were prepared for histological evaluation.

### 2.2. RNA Extraction and Transcriptome Library Preparation

In total, seven tumor and matched untreated tumor samples were eligible for sequencing analysis. Total RNA was extracted from 10 μm-thick DNase-treated formalin fixated, paraffin embedded (FFPE) tissue sections prepared by microtome using the Maxwell RSC FFPE RNA kit (Promega Corporation, Madison, WI, USA). Extracted RNA was quantified by Qubit 4.0 using the RNA HS Assay Kit (Thermo Fisher Scientific Inc., Waltham, MA, USA). A maximum of 50 ng of extracted RNA was used to prepare cDNA libraries after fragmentation of total RNA at 94 °C for 3 min. Genome-wide transcriptome sequencing libraries were prepared using the commercially available Takara Smarter total-RNA Seq kit V2.5 Pico Input Mammalian (Takara Bio Inc., Kusato, Japan) [15]. In brief, cDNA libraries were prepared by modified random N6 hexamer reverse oligonucleotides. During the first PCR round of amplification, full-length Illumina adapters, including barcodes sequences, were added. Afterwards, ribosomal cDNA sequences were depleted in the presence of ribonuclease (RNase H) and the mammalian-specific R-Probes. The remaining cDNA fragments were enriched by a second round of amplification using universal primer oligos specific to all adapter sequences. The final library contained sequences allowing clustering on most common Illumina flow cells. To equalize the amount of input material from each sample, cDNA libraries were quantified by spectrophotometer Qubit 4.0 DNA HS Assay Kit (Thermo Fisher Scientific Inc., Waltham, MA, USA) [11,16]. In addition, the quality control on cDNA libraries of each sample was performed using Ag-ilent 2100 Bioanalyzer (Agilent Technologies, Inc., Santa Clara, CA, USA).

### 2.3. Illumina Sequencing and Gene Expression Analysis

Sample RNA-seq libraries were normalized and sequenced on a NextSeq 500 Illumina instrument (Illumina, Inc., San Diego, CA, USA). All sequencing was done on mid-output V2.5 flow cell Kit, to generate a median of 25 million raw paired-end reads (2 × 75) for each cDNA sample. Takara indices, with TruSeq 96 CD Illumina adapter dual indexes, were used in sample sheet to demultiplex and assign specific sequence reads. Collected gene expression data were analyzed using bioinformatic toolbox at Chipster virtual interface at IT Center for Science (CSC), Finland [17]. In brief, adapters were trimmed, and sequencing reads were preprocessed. The sequencing data were then checked for quality using FastQC. Paired-end sequencing reads were mapped to Homo sapiens genome version release GRCh38.95 with splice aware aligner STAR [18]. BAM files were then used as input files to quantitate the number of short reads with HTSeq, which resulted in the aligned read counts for all sequenced gene transcripts [19]. Differential expression analysis was performed using the DESeq2 Bioconductor package. Briefly, normalized control and treatment count tables were merged and were used as a template for differential expression analysis and to generate fold change values in the log2 scale. Genes with an adjusted *p* ≤ 0.05 and log2fold change of +1 or −1 were considered as significantly expressed. Hierarchical clustering heatmaps and dendrograms of RNA expression profiles of 7 cases were generated using DESeq2. Overrepresentation analysis of differentially expressed genes was completed by ConsensusPathDB, which contains pathway information from over 30 publicly available databases [20].

### 2.4. PDAC Cell Line Culturing Condition

Pancreatic adenocarcinoma cell line HPAF-II (ATCC: CRL-1997) was obtained from American Type Culture Collection (ATCC, Manassas, VA, USA), and pancreatic carcinoma cell line PANC-1 (DSMZ: ACC-no. 783) was obtained from Leibniz Institute (DSMZ, Braunschweig, Germany). Both cell lines were grown in Eagle’s Minimum Essential Medium (EMEM) (30-2003, ATCC, Manassas, VA, USA) with 10% fetal bovine serum (FBS) (Thermo Fisher Scientific Inc.) at 37 °C in a humidified chamber with 5% CO_2_ supplement. HPAF-II (600 cells/mm^2^) and PANC-1 (400 cells/mm^2^) cultivated in a complete growth medium for 24 h. To measure cell viability, after 24 h, the cells were treated with 200 μM of IPA (Sigma-Aldrich Chemical Company, St. Louis, MI, USA) for 24, 48, and 72 h in a 96-well plate. To measure mRNA expression, IPA treatment with a concentration of 200 μM was carried out for 6, 12, and 24 h in 6-well plates in triplicates. Untreated cells served as a control at different time points.

### 2.5. Cell Viability Assay

Cell-titer Glo-luminescent cell viability assay (Promega Corporation, Madison, WI, USA) was used to determine the cell proliferation/cytotoxicity. Briefly, cells were seeded at the indicated density for 24 h in 96-well plate, and media were replaced and subsequently treated with 200 µM of IPA for 24, 48, and 72 h. At termination of exposure, media were replaced with fresh media and measured using the luminescence-based CellTiter-Glo^®^ 2.0 cell viability assay kit (Promega Corporation, Madison, WI, USA), in accordance with the instructions of the manufacturer, and luminescence was measured using CLARIOstar^®^ FLx100 Luminometer (BMG Labtech, Ortenbery, Germany).

### 2.6. RNA Extraction and Two-Step Real-Time PCR

Total-RNA was isolated using Maxwell RSC simplyRNA Tissue Kit (AS1340, Promega Corporation, Madison, WI, USA) by Maxwell RSC Instrument (Promega Corporation, Madison, WI, USA), in accordance with the instructions of the manufacturer. cDNAs were generated from a maximum of 2000 ng of total RNA by High-Capacity RNA-to-cDNA Kit (4388950, Thermo Fisher Scientific Inc.), in accordance with the instructions of the manufacturer. Real-time PCR was performed from 1/8-diluted cDNA using TaqMan Fast Advanced Master Mix (4444965, Thermo Fisher Scientific Inc.). Custom TaqMan Gene Expression Assays were used for primer targeting TiPARP (assay ID: Hs00296054_m1), CYP1A1 (assay ID: Hs010554794_m1), CYP1B1 (assay ID: Hs90164383_m1), and GAPDH (assay ID: Hs02758991_g1). All PCR amplification reactions were run in triplicates on a QuantStudio 3 Realtime PCR System (Thermo Fisher Scientific Inc.).

### 2.7. Statistical Evaluation

Cell line treatments were performed in at least three biological replicates. For investigating cell viability, the data were normalized to control (untreated) at different time points and expressed as cell viability (percentage of control). Statistical analysis was carried out using an unpaired Student’s *t*-test. A *p*-value of 0.05 was set as the threshold for statistical significance. Data were analyzed with GraphPad Prism software, version 9.0.0 (GraphPad Software Inc., San Diego, CA, USA).

In terms of real time PCR, the relative expression was calculated by the 2^−ΔΔCt^ [21], applying GAPDH as the reference gene, and normalized to the average ΔCt value of untreated control cells for different time points.

## 3. Results

### 3.1. Effect of IPA on Ex Vivo Tissue Culture

#### 3.1.1. RNA Expression

Differential expression analysis using the DESeq2 identified significant (adjusted *p* ≤ 0.05) upregulation of seven genes (0.013%, out of 53,688 nonzero total read count) after treatment with 200 μM of IPA for 48 h (Figure 1).

Overrepresentation analysis of the differentially expressed genes suggested strong correlation to the aryl hydrocarbon receptor (AHR) pathway counting cytochrome P450 1A1 (CYP1A1), cytochrome P450 1B1 (CYP1B1) genes, and AHRR (aryl hydrocarbon receptor repressor) as key indicators. The most significantly associated set of genes in pathways are shown in Table 2.

Furthermore, transcripts of the genes CYP1B1-AS1 (CYP1B1 Antisense RNA 1), TiPARP ((TCDD)-inducible poly (ADP-ribose) polymerase), NKD2 (Naked cuticle 2), and AC015712.2 (long non-coding RNA lncRNA) were upregulated.

#### 3.1.2. Pathological Examination of FFPE Tissue

Histopathological examination on Hematoxylin and Eosin (H&E) staining revealed no obvious signs of IPA-treatment-related cell damage, neither in the tumor nor in the non-tumorous tissue components (stroma and remnants of pancreatic parenchyma) as shown in Figure 2.

### 3.2. Effect of IPA on Pancreatic Cell Lines

The effect of IPA on cell growth at 24, 48, and 72 h was evaluated in the two PDAC cell lines, HPAF-II and PANC-1. As shown in Figure 3, neither HPAF-II nor PANC-1 cells showed a decrease in viable cell number compared to control cells after 24, 48, and 72 h of treatment. No significant cytotoxic effects were observed at 200 µM of IPA in any of cell lines (Figure 3).

In addition to the effect on cell growth, the mRNA expression of TiPARP, CYP1A1, and CYP1B1 in HPAF-II and PANC-1 after being treated with IPA (200 µM) after 6, 12, and 24 h were examined (Figure 4). Compared to the control, IPA treatment increased TiPARP expression 2.4-fold for HPAF-II and 3.5-fold for PANC-1 after only 6 h. IPA treatment increased the TiPARP expression in a time-dependent manner in HPAF-II cells This time-dependent activation is not observed in PANC-1, but after 24 h the IPA treatment still increased the TiPARP mRNA level by 3.4 times compared to the control (untreated cells).

Additionally, mRNA expression of CYP1A1 and CYP1B1 was also increased in a time-dependent manner for any of cell lines with IPA treatment compared to the untreated control (Figure 4).

## 4. Discussion

In this study, we demonstrate that a PDAC ex vivo tissue slice culture approach can adequately respond to a selected exogenous substance. The model we created reflects the complex microenvironment of the tumor including all cell types in the same 3D architecture as the patient’s tumor. However, evaluating the efficacy of new therapeutic strategies requires robust preclinical models that accurately reflect the cellular biology of patients’ diseases, so we characterized our model by examining the effect of IPA on the AHR signaling pathway.

IPA is actually an aromatic pyruvic acid that has antioxidant properties, it was able to reduce oxidative cell damage in the brain, and the severity of skin lesions caused by UVB in mice. As a keto analog of tryptophan, it participates in the biosynthesis of the plant hormone auxin, thus it is precursor of indole-3-acetaldehyde, indole-3-aldehyde, and indole-3-acetic acid. In addition to the fact that plants, fungi and some microorganisms are able to produce IPA by deamination of tryptophan, the compound can also be synthesized by the action of aromatic amino acid transaminases. Its well-known property is that it can activate the AHR signal pathway and acts as an AHR proagonist [13].

Tryptophan metabolites, such as IPA and aromatic hydrocarbons metabolites, including 2,3,7,8-tetrachlorodibenzo-p-dioxin (TCDD), and benzo(a)pyrene) metabolites, can activate the AHR signaling pathway. As presented in Figure 5, unbound AHR can form a complex with two heat shock proteins (HSP90), an X-associated protein (AIP), and a p23 molecular chaperone protein in the cytosol. Binding of the ligand activates the AHR complex, so it moves into the nucleus and detaches from the chaperone subunits to dimerize with the aryl hydrocarbon receptor nuclear translocator (ARNT). In the nucleus, the ARNT–AHR complex binds to xenobiotic response elements (XRE) and can transactivate target genes in the promoter regions, where AHR indeed regulates the expression of several groups of genes in the genome, such as the cytochrome P450 family group CYP1A1, CYP1B1, AHRR, and TiPARP [11,22].

In our study, tissues exposed to IPA treatment and untreated, comparable controls were analyzed using genome-wide RNA sequencing to evaluate the effect of IPA on gene expression. We confirmed that IPA is an inducer of the CYP1A1, CYP1B1, AHRR, and TiPARP genes in the patient-derived tissue slices model, as the overrepresentation and clustering of upregulated genes showed that the AHR pathway was indeed significantly induced. The effect of IPA on TiPARP, CYP1A1, and CYP1B1 mRNA expression was also validated in in vitro conditions. Our result was consistent with the experiment of Reiji Aoki et al., who demonstrated that AHR genes are significantly upregulated in the intestinal tract of mice both in vitro and in vivo [13].

Similar to other signaling pathways, the AHR signal also contains a negative feedback loop, the presence of which we also found during our experiments. As the cell strives to maintain homeostasis, the signal to the AHR can be attenuated by another protein, the AHR repressor (AHRR), with expression that increases rapidly upon AHR activation. It is structurally like the AHR, but contains a strong transcriptional repressor domain and does not require an agonist to dimerize with ARNT. Another level of negative regulation is mediated by prototypical target genes, such as AHR induction by CYP1A1 and CYP1B1, which repress the AHR through the metabolism of AHR regulatory ligands [24,25,26].

Through its target genes, the AHR is involved in many regulatory processes, including xenobiotic metabolism, vascular development, immune responses, and cell cycle control. During our study, we observed an increase in TiPARP mRNA expression, which can be considered one of its target genes, in both ex vivo and in vitro conditions. Recently, the results of Giulia Grimaldi et al. indicated that TiPARP also negatively regulates AHR, revealing a novel negative feedback loop, where increased levels of TiPARP suppress AHR activity in a way that requires TiPARP catalytic activity [23]. Among the clustered genes, we also observed over-expression of the non-coding CYP1B1-AS1 antisense RNA 1. CYP1B1-AS1 is a non-annotated gene which by far has not clearly been linked to any specific pathway. This transcript has probably an intermediary role in establishing interactions with other splicing components involved in pre-messenger RNA (pre-mRNA) processing.

In conclusion, our study was the first to reveal the induction of the AHR signaling pathway by IPA in an ex vivo tissue explant model of human pancreatic cancer. Our study clarifies that AHR signaling genes are detected and can be quantified at the transcriptome level. We also conclude that the activation of AHR genes does not have a visible toxic effect on the PDAC tissue. In contrast, we did not study the tissue and cell line at the protein expression level, nor did we investigate the spatial gene expression pattern. However, our results demonstrate that, through the induction of the molecular signaling pathway, our model can produce consistent results in vitro, while the experimental model system most ideally reflects the tissue structure and biology of a patient’s tumor. However, like all models, this one also has its limitations, since we only know the experimental system from the tissue samples of operable patients. Thus, although we do not believe that this model can replace the need for mouse models, we recommend that the ex vivo tissue slice model be considered as a clinically relevant and reliable addition to the currently available preclinical models during drug-development processes.

## Figures and Tables

**Figure 1 antioxidants-12-00167-f001:**
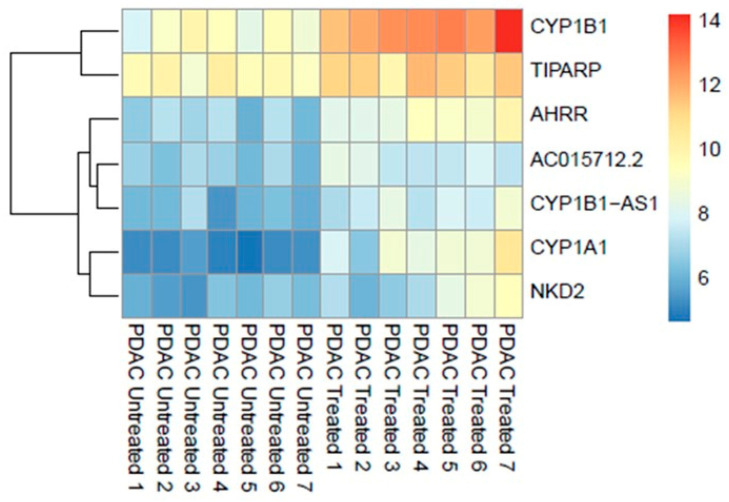
Genome-wide transcriptomic heatmap and dendrogram of the differentially expressed genes in PDAC ex vivo tissues after treatment with IPA (200 μM) at 48 h.

**Figure 2 antioxidants-12-00167-f002:**
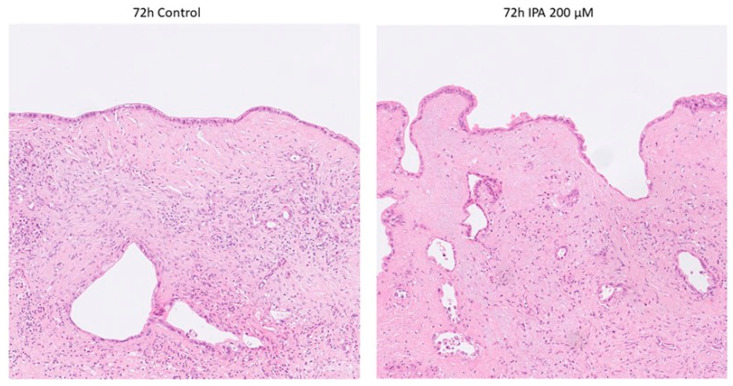
Representative photomicrographs of control and IPA-treated ex vivo cultured tissue slices. None of the tumor specimens showed any sign of treatment-related cell damage. Non-tumorous tissue components remained well preserved.

**Figure 3 antioxidants-12-00167-f003:**
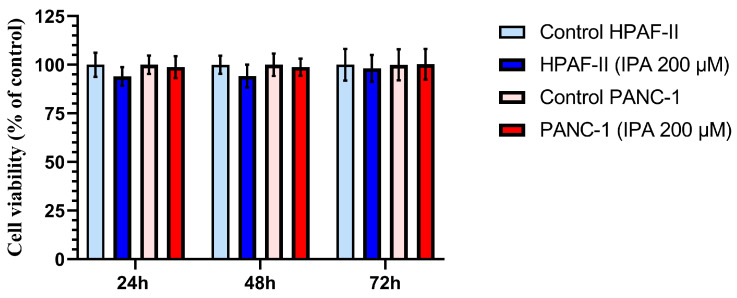
Effect of IPA treatment on the cell viability of HPAF-II and PANC-1 compared to control cells at different time points. Pancreatic cancer cell lines, PANC-1 and HPAF-II, were treated with IPA 200 μM for 24, 48, and 72 h. Cell viability is shown in the graph and normalized to the mean value of the control cells, and data are represented as mean ± SD (*n* = 3). The statistical analysis is performed using an unpaired Student’s *t*-test, and the analysis shows no significant difference between the treated (IPA 200 μM) and control cells.

**Figure 4 antioxidants-12-00167-f004:**
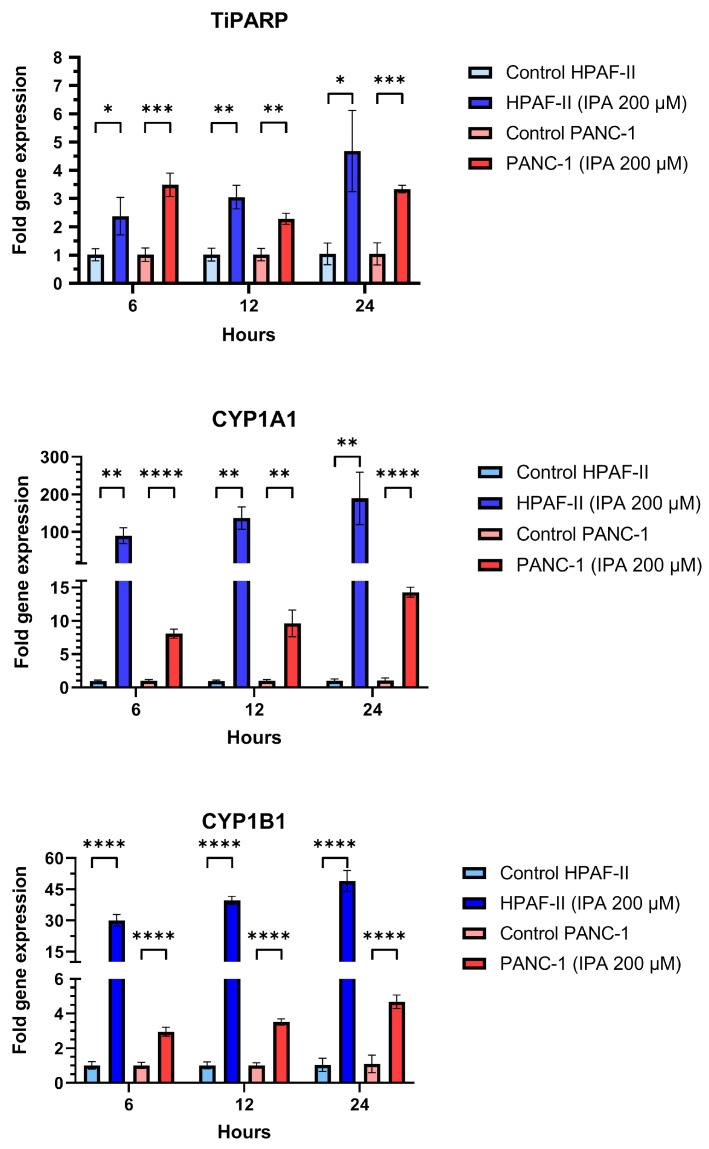
Effect of IPA treatment on mRNA expression of TiPARP, CYP1A1, and CYP1B1. PANC-1 and HPAF-II were treated with IPA for 6, 12, and 24 h and the mRNA expression of TiPARP, CYP1A1, and CYP1B1 was determined from the collected RNA of each sample using qPCR. The graphs represent the relative expression of each gene using the formula 2^−ΔΔCt^ normalized to the mean ΔCt of control cells. Data are represented as mean ± SD (*n* = 3). Statistical comparison between IPA-treated and untreated (control) cell lines were performed using an unpaired Student’s *t*-test and the significant differences are indicated by asterisks (**** *p* ≤ 0.0001, *** *p* ≤ 0.001, ** *p* ≤ 0.01, * *p* ≤ 0.05).

**Figure 5 antioxidants-12-00167-f005:**
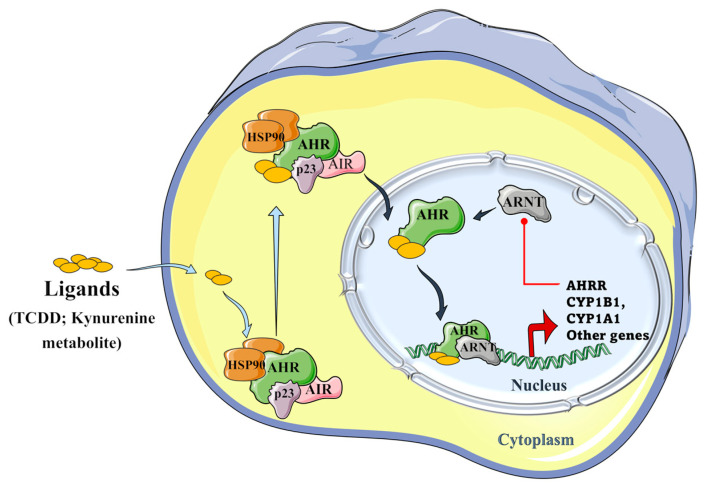
Schematic representation of the aryl hydrocarbon receptor (AHR) signaling pathway. The AHR can form a complex with its ligands (TCCD and kynurenine metabolite), the two heat shock proteins, i.e., HSP90s; with an X-associated protein (AIP), and a molecular chaperone protein, p23 in the cytoplasm. The formed AHR complex is then able to translocate into the nucleus, where it detaches from chaperone subunits, in order to dimerize with the aryl hydrocarbon receptor nuclear translocator (ARNT), and to transactivate the target genes by binding to xenobiotic response elements (XRE) in their promoter regions, where AHR regulates the expression of several groups of genes in the genome, such as CYP1A1, CYP1B1, AHRR, and TiPARP genes. The induced aryl hydrocarbon receptor repressor (AHRR) plays an important role in a negative regulatory step because it can inhibit the binding of ARNT and AHR and, hence, further transactivation [22,23,24,25] (Figure 5 was partly generated using Servier Medical Art, provided by Servier, licensed under a Creative Commons Attribution 3.0 unported license).

**Table 1 antioxidants-12-00167-t001:** Clinicopathological data of the study samples.

Culture ID	Gender	Preoperative Chemotherapy	Histological Type	Grade of Differentiation	Stage *
1	Female	No	PDAC	Moderate	pT3 N2
2	Male	No	PDAC	Poor	pT2 N1
3	Male	No	PDAC	Moderate	pT2 N0
4	Female	No	PDAC	Poor	pT3 N2
5	Male	No	PDAC	Moderate–poor	pT3 N2
6	Male	No	PDAC	Moderate	pT3 N2
7	Female	No	PDAC	Moderate–poor	pT3 N2

* Stage–TNM classification (8th Edition).

**Table 2 antioxidants-12-00167-t002:** Most significant pathways are shown. Additionally, CYP1A1 and CYP1B1 are significantly involved in many other ConsensusPathDB-detected signalings (Most significant data is shown in the table below).

*p*-Value	Pathway	Database	Genes Mapped
1.66 × 10^−7^	Aryl Hydrocarbon Receptor	Wikipathways	AHRR; CYP1A1; CYP1B1
2.13 × 10^−7^	Amodiaquine Pathway Pharmacokinetics	PharmGKB	CYP1A1; CYP1B1

## Data Availability

Not applicable.

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
