# Peer review of "Exploration of Patient-Derived Pancreatic Ductal Adenocarcinoma Ex Vivo Tissue for Treatment Response"

_antioxidants, 2023, doi:10.3390/antiox12010167_

Round 1

Reviewer 1 Report

This is an interesting work on the pancreatic ductal adenocarcinoma based on a novel ex vivo culture model. I do not have particular major observations for this work that can be accepted in antioxidants following several minor improvements. Please see below several minor suggestions for improving the work.
1)    For a better understanding of the work purposes, I suggest including the word “indole-3-pyruvic acid” in the title
2)    Line 19 it can be “p>0.05”. The p value should also be included in the following sentence as the authors stated “significant”
3)    I suggest including a few sentences detailing the current PDAC incidence, 5 years survival rate and currently known risk factors in the introduction. Please also briefly mention the currently employed antitumor therapies against PDAC doi: 10.3390/diagnostics11030562.
4)    Please explain in the introduction why did the authors selected indole-3-pyruvic acid for this experimental design
5)    Sections .2.2 and 2.7 are without references
6)    Why did the authors selected 200 μM as treatment concentration?
7)    Line “ine 200 “lines. (Figure 3)”. “(Figure 3)” should be before the period. I have noted several similar typo errors in other sections of the manuscript. Please check it entirely
8)    Fig 4 should depict statistical comparison and relative p values, if differences being significant
9)    Lines 230-231, please include this additional reference on P450 family genes (doi: 10.1002/etc.4029)
10)    The figure 4 of the discussion should be figure 5
11)    Line 273, please include this additional reference on AHR genes (PMID: 30003042)
12)    Please include study limitations in the discussion. For instance there is no evaluation of protein levels of selected AHR genes. This can be done by western blot in cultured cells treated with indole-3-pyruvic acid

Author Response

Dear Reviewer 1,

Together with my co-authors, we would like to thank you for your comments on our manuscript and we are grateful for your work on our manuscript, because we feel that your comments have deservedly improved the quality of our manuscript. You can find our answers in the attached PDF. 

Yours sincerely,

Tímea Szekerczés Ph.D.

Reviewer 2 Report

Indeed, resected tumor sections cultured ex vivo have recently attracted interest in precision medicine because they reflect the complex microenvironment of cancer tissue. In this study, the authors examined the treatment response of an in-house ex vivo pancreatic ductal adenocarcinoma (PDAC) tissue culture model and in vitro assay. This study provides evidence that ex vivo patient tissue models are a valuable tool in precision medicine for monitoring the effect of personalized treatment.

Why was indole-3-pyruvic acid chosen?

The authors do not provide any other characteristics of the tumor other than histology. There is no information whether the response to treatment depends on the initial molecular characteristics, degree of differentiation, stage, etc.

Small remarks. In the Discussion section, you need to replace Figure 4 with Figure 5.

Author Response

Dear Reviewer 2,

Together with my co-authors, we would like to thank you for your comments on our manuscript and we are grateful for your work on our manuscript, because we feel that your comments have deservedly improved the quality of our manuscript. You can find our answers in the attached PDF.

Yours sincerely,

Tímea Szekerczés Ph.D.
